# Ocular-Component-Specific miRNA Expression in a Murine Model of Lens-Induced Myopia

**DOI:** 10.3390/ijms20153629

**Published:** 2019-07-24

**Authors:** Yasuhisa Tanaka, Toshihide Kurihara, Yumi Hagiwara, Shin-ichi Ikeda, Kiwako Mori, Xiaoyan Jiang, Hidemasa Torii, Kazuo Tsubota

**Affiliations:** 1Department of Ophthalmology, School of Medicine, Keio University, Shinjuku-ku, Tokyo 160-8582, Japan; 2Laboratory of Photobiology, School of Medicine, Keio University, Shinjuku-ku, Tokyo 160-8582, Japan; 3Santen Pharmaceutical Co., Ltd., Osaka 530-8582, Japan

**Keywords:** miRNA, lens-induced myopia, ocular components

## Abstract

To identify tissues and molecules involved in refractive myopic shift and axial length elongation in a murine lens-induced myopia model, we performed a comprehensive analysis of microRNA (miRNA) expression. Three weeks after negative 30 diopter lens fixation on three-week-old C57BL/6J mice, total RNA was extracted from individual ocular components including cornea, iris, lens, retina, retinal pigment epithelium (RPE)/choroid, and sclera tissue. The miRNA expression analysis was pooled from three samples and carried out using Agilent Mouse miRNA Microarray (8 × 60 K) miRBase21.0. The expression ratio was calculated, and differentially expressed miRNAs were extracted, using GeneSpring GX 14.5. Myopic induction showed a significant myopic refractive change, axial elongation, and choroidal thinning. Through the comprehensive miRNA analysis, several upregulated miRNAs (56 in cornea tissue, 13 in iris tissue, 6 in lens tissue, 0 in retina tissue, 29 in RPE/choroid tissue, and 30 in sclera tissue) and downregulated miRNAs (7 in cornea tissue, 28 in iris tissue, 17 in lens tissue, 9 in retina tissue, 7 in RPE/choroid tissue, and 40 in sclera tissue) were observed. Overlapping expression changes in miRNAs were also found in different ocular components. Some of this miRNA dysregulation may be functionally involved in refractive myopia shift and axial length elongation.

## 1. Introduction

In 2000, there were 1.4 billion myopic people globally without correction of refractive errors [1]. It has been predicted that the population of people with myopia will increase to 4.8 billion by 2050 [1,2]. This expansion in the prevalence of myopia is dramatic; therefore, there is an urgent need to understand the mechanisms underlying the development and progression of myopia and establish a treatment. Previous studies have revealed some of the molecular and cellular mechanisms that underlie myopia development; for example, a large genome-wide analysis and a meta-analysis have been performed [3,4]. However, more data are needed to identify the causative factors and biomarkers of myopia progression.

As experimental animal models of myopia, two types of induction, form-deprived myopia (FDM) and lens-induced myopia (LIM), have been performed in mice [5], chicks [6,7], guinea pigs [8,9], tree shrews [10], rabbits [11], and monkeys [12]. Proteomic and microarray analyses have been performed using these experimental animal models of myopia, and changes in several factors were reported [13,14].

MicroRNAs (miRNAs) are single-stranded RNAs comprised of non-coding RNAs of 21–24 nucleotides in length that are known to inhibit gene expression after transcription [15]. MiRNAs play an important role in various diseases [16,17], and may be used as biomarkers of disease. MiRNAs may exist in serum and plasma [18,19], and miRNAs from validated blood samples can be used as predictors of myopia progression. Several previous studies focused on miRNAs involved in myopia progression suggested the possibility of using miRNAs as biomarkers of myopia progression [20,21].

In this study, using a murine model of LIM [5], a comprehensive analysis of miRNA was performed in individual ocular tissues, including cornea, iris, lens, retina, retinal pigment epithelium (RPE)/choroid, and sclera tissues, with the aim of investigating the expression pattern of and change in miRNAs during myopia progression.

## 2. Results

### 2.1. Refractive Error and Axial Length Changes in the Murine Model of Myopia

Three-week-old C57/BL/6J mice wore a negative 30 diopter (D) lens on the right eye. After three weeks, the changes in refractive error and axial length were evaluated. The naïve right eyes of the mice that were not wearing a lens were used as normal eyes. The eyes with lens-induced myopia showed a significant myopic shift in refractive error (−18.42 D ± 3.98) compared to normal eyes (0.95 D ± 1.85) (*p* < 0.001, Figure 1a). A significant axial elongation was also observed in eyes with lens-induced myopia (0.273 mm ± 0.009) compared to normal eyes (0.155 mm ± 0.015) (*p* < 0.001, Figure 1b).

### 2.2. Change in Retinal and Choroidal Thickness

Three-week-old C57/BL/6J mice wore a negative 30 D lens on the right eye. After three weeks, the changes in retinal and choroidal thickness were evaluated. The naïve right eyes of the mice that were not wearing a lens were used as normal eyes. The eyes with lens-induced myopia showed a reduction in retinal thickness (−12.432 μm ± 8.937) compared to normal eyes (−1.887 μm ± 10.417) (Figure 2a). A significantly thinner choroid was also observed in eyes with lens-induced myopia (–4.196 μm ± 1.716) compared to normal eyes (0.405 μm ± 0.995) (*p* < 0.05, Figure 2b).

### 2.3. Differentially Expressed miRNAs in Individual Ocular Components During Myopia Progression

Each eyeball was separated into cornea, iris, lens, retina, RPE/choroid, and sclera (Figure 3). The miRNA array was evaluated using the tissues of the individual ocular components. A number of upregulated miRNAs (56 in cornea tissue, 13 in iris tissue, 6 in lens tissue, 0 in retina tissue, 29 in RPE/choroid tissue, and 30 in sclera tissue) and downregulated miRNAs (7 in cornea tissue, 28 in iris tissue, 17 in lens tissue, 9 in retina tissue, 7 in RPE/choroid tissue, and 40 in sclera tissue) were found in myopic tissues compared to normal tissues (Table 1, Figure 4, Figure 5, Figure 6, Figure 7, Figure 8 and Figure 9).

### 2.4. Overlapping Expression Changes in miRNAs in Different Ocular Component Tissues

Overlapping expression changes in miRNAs were found in different ocular tissues. Table 2 shows the miRNAs that were upregulated in both corneal and other tissues (4 in iris tissue, 2 in lens tissue, 3 in RPE/choroid tissue, and 15 in sclera tissue), the miRNAs that were upregulated in corneal tissue and downregulated in other tissues (three in iris tissue and two in sclera tissue), the miRNAs that were downregulated in corneal tissue and upregulated in other tissues (two in RPE/choroid tissue), and the miRNAs that were downregulated in both corneal tissue and other tissues (one in iris tissue and three in lens tissue). Table 3 shows the miRNAs that were upregulated in both iris and other tissues (four in RPE/choroid tissue and one in sclera tissue), the miRNAs that were upregulated in iris tissue and downregulated in other tissues (one in lens tissue and three in sclera tissue), the miRNAs that were downregulated in iris tissue and upregulated in other tissues (six in RPE/choroid tissue and one in sclera tissue), and the miRNAs that were downregulated in both iris tissue and other tissues (one in lens tissue, one in RPE/choroid tissue, and four in sclera tissue). Table 4 shows the one miRNA that was upregulated in both lens and RPE/choroid tissues, the three miRNAs that were upregulated in lens tissue and downregulated in sclera tissue, the miRNAs that were downregulated in lens tissue and upregulated in other tissues (one in RPE/choroid tissue and one in sclera tissue), and the miRNAs that were downregulated in both lens and other tissues (two in retina tissue and two in sclera tissue). Table 5 shows the two miRNAs that were downregulated in retina tissue and upregulated in RPE/choroid tissue. Table 6 shows the four miRNAs that were upregulated in both RPE/choroid tissue and sclera tissue, the two miRNAs that were upregulated in RPE/choroid tissue and downregulated in sclera tissue, and the two miRNAs that were downregulated in RPE/choroid tissue and upregulated in sclera tissue. In these expression changes in miRNAs, 18 miRNAs overlapped in more than three different types of ocular component tissue (Table 7).

### 2.5. Affected miRNAs and Their Predicted Target mRNA in Different Ocular Component Tissues

Based on the myopia-induced changes in miRNA expression, the target mRNA was predicted using MiRTarBase. Table 8, Table 9, Table 10, Table 11 and Table 12 show the target genes predicted from the change in miRNA expression in the cornea (73 genes from 20 miRNAs), the iris (27 genes from 11 miRNAs), the lens (32 genes from 8 miRNAs), the retina (no detection), the RPE/choroid (22 genes from 5 miRNAs) and the sclera (89 genes from 21 miRNAs). The genes shown in bold were overlapped in each ocular tissue. Overlapping expression changes in miRNAs were found in two and three types of ocular tissue, and their target genes are shown in Table 13 and Table 14.

## 3. Discussion

In this study, we performed a comprehensive miRNA analysis of ocular component tissues, including cornea, iris, lens, retina, RPE/choroid, and sclera tissues, from an experimental murine model of myopia. A number of differentially expressed miRNAs were observed in each ocular component. Overlapping expression changes were also found in different ocular components.

Several mRNA expression changes in FDM or LIM mice were previously reported, including an increase in *fibroblast growth factor* (*FGF*) *10* expression in sclera tissue [22] and a decrease in *Wingless* (*WNT*) *2b*/*Frizzled* (*FZD*) *5*/*β-catenin* expression in retina tissue [23]. An increase in *Wnt3*/*b-catenin* gene expression [24] and a decrease in *TGF-β*/*Col1* gene expression [9] were also reported in FDM guinea pigs. Furthermore, previous studies showed that various gene expression changes may be functionally related to myopic phenotypes. A myopic shift in refractive errors and elongation of the axial length were reported in *EGR1*-deficient [25], *M2*-deficient [26], *LRP2*-overexpressed [27], *APLP2*-deficent [28] and *Lumican*-overexpressed [29] mice. Furthermore, studies in human cohorts have identified a number of genes related to myopia development [30]. In our study, a range of target mRNAs were predicted from expression changes in miRNA expression. The predicted genes were found to correspond to *EGR-1* and *TGF*-*β* or be similar to *FGF*, *WNT*, and *FZD*, which were described in the abovementioned studies. Although the predicted genes were different between myopic induction models and species, these genes may play an important role in myopia progression. On the other hand, other genes, such as *PTEN* in the cornea and *VSIVGP2*, *NOTCH1*, *STAT3*, and *CLICS* in the sclera, were found to not correspond to previous reports, suggesting that these genes may have a function that is specific to each ocular component.

It has been reported that miR-200a/b/c expression overlaps in a range of tissues with a tubular structure, including kidney tissue (proximal tubule and collecting duct), lung tissue, pancreas tissue (duct cells), small intestine tissue (intestinal villus), bile duct tissue, and exocrine gland tissue (duct cells). Furthermore, miR-200a/b/c expression was found to be increased in plasma from the site of an acute kidney injury, suggesting that miR-200a/b/c may be used as a biomarker for kidney and other tubular structure organ injury [31]. In the current study, we found overlapping changes in miRNA expression in two and three types of ocular tissue (Table 2, Table 3, Table 4, Table 5, Table 6 and Table 7). These individual ocular components are in close proximity to and functionally connected with each other. Thus, we suggest that overlapping changes in miRNA expression among different ocular components can be used as myopic diagnosis markers.

In a previous study, eight miRNAs were found to be upregulated and to overlap with retinas and whole eyeballs in a murine model of form-deprivation myopia. The authors screened out 1805 target genes for the eight differentially expressed miRNAs, including *MAPK-10* [32]. In the present study, we also found a number of overlapping miRNAs in individual ocular components together with predicted target genes (Table 13 and Table 14). Although these new target genes were found to not exactly correspond to previous reports, we speculate that these genes may be important factors in the suppression or acceleration of myopia progression.

Comprehensive approaches to the analysis of mRNA and miRNA expression have also been reported. In sclera from an FDM mice model, Let-7a, miR-16-2, *Smok4a*, *Prph2*, and *Gnat1* expression were found to fluctuate [14]. Fifty-three (53) miRNAs were previously reported to be either upregulated or downregulated in the retina of LIM mice, and mmu-miR-671-5p was identified among them [33]. In this study, 18 miRNAs were identified as being differentially expressed in three different ocular components. These miRNAs may play an important role in myopia progression. Mmu-miR-7047-3p, mmu-miR-7085-3p, and mmu-miR-96-5p showed an opposite change in expression between the anterior (cornea, iris, and lens) and posterior (retina, RPE/choroid, and sclera) ocular components. Although some differences exist between animal species [34], the expression and functions of miRNAs are largely evolutionarily conserved [35]. In the present study, we identified both discrete and overlapping changes in miRNA expression in individual ocular components during myopia progression in a murine model. Further studies will be conducted to explore miRNA profiles in order to understand the molecular pathogenesis of human myopia progression in humans and to establish of biomarkers for its prediction.

## 4. Materials and Methods

### 4.1. Experimental Animals

The experimental protocol used in this study complied with the National Institutes of Health (NIH) guidelines for working with laboratory animals, the ARVO Statement for the Use of Animals in Ophthalmic and Vision Research, and the Animal Research: Reporting of In Vivo Experiments (ARRIVE) guidelines. The experimental protocol was approved by the Institutional Animal Care and Use Committee at Keio University. C57BL/6J male mice (CLEA Japan, Yokohama, Japan) were maintained by free intake of a standard diet (MF, Oriental Yeast Co., Ltd., Tokyo, Japan) and water, with three mice in one cage. The mice were raised in an environment with a 12 h/12 h light/dark cycle (the dark cycle from 8:00 p.m. to 8:00 a.m.) at 23 ± 3 °C. The light cycle was maintained using a 50-lux background. These conditions were based on a previously reported experimental murine model of myopia [5]. The animal trial was approved by the ethics committee of Keio University (ethics review number: 16017-(1), 25 October 2017).

### 4.2. Myopia Induction

Before and after the myopia induction, the refraction and axial length of all eyes were measured using a refractometer (Steinberis Transfer Center, Tübingen, Germany) and spectral domain optical coherent tomography (SD-OCT, Envisu R4310, Leica, Wetzlar, Germany), respectively, under anesthesia by medetomidine (0.75 mg/kg, Sandoz K.K., Tokyo, Japan), midazolam (4 mg/kg, Domitor^®^, Orion Corporation, Espoo, Finland), and butorphanol tartrate (5 mg/kg, Meiji Seika Pharma Co., Ltd., Tokyo, Japan) dissolved in normal saline (MMB). For the myopia induction group, a −30 D lens was fixed onto the right eye at postnatal three weeks old, and the mice were kept for three weeks. For the normal group, mice were prepared and kept without any special treatment. The induction of myopia and the ocular measurement were based on a previously reported experimental murine model of myopia [5]. Three mice were used for the myopia induction group and the normal group, respectively. In accordance with a previous report [36], the thickness of the retina and choroid was captured at points that were ±300 μm and ±400 μm from the optic nerve, respectively, and measured using the NIH ImageJ software. 

### 4.3. miRNA Extraction

After euthanasia was performed by intraperitoneal MMB injection of an overdose of anesthesia, right eyes from the myopia induction group and right eyes from the control group were enucleated and separated into cornea, iris, lens, retina, RPE/choroid, and sclera tissues. The separated ocular tissues were put into QIAzol and homogenized. Total RNA extraction was performed using a miRNeasy Micro kit (QIAGEN, Venlo, Netherlands) according to the manufacturer’s instructions.

### 4.4. miRNA Microarray

After mixing equal amounts of three samples extracted from the same eye tissues, a template of 100 ng total RNA was applied to Agilent Mouse miRNA Microarray (8 × 60 K) miRBase 21.0 (Agilent, Santa Clara, CA, USA). Cyamin-3 labeling samples were hybridized at 55 °C and 20 rpm for 17 h using a miRNA Complete Labeling and Hyb Kit (Agilent) and an Expression Hybridization Kit (Agilent). After hybridization, the microarrays were scanned using a DNA Microarray Scanner (Agilent). The scanning data were digitized using the Feature Extraction version 10.7.1.1 software (Agilent). The miRNA microarray protocols were applied by following each manufacturer’s instructions. The microarray datasets are displayed in the National Center for Biotechnology Information (NCBI) Gene Expression Omnibus with the accession number GSE131831.

### 4.5. Data Analysis

miRNA microarray data were analyzed using the Geneview Data module in GeneSpring GX 14.5 (Agilent). The expression ratio of sample molecules for the denominator was calculated according to the combination described in Table 1. A fold-change of >2 and <0.5 with a detected flag in both the denominator and the numerator or a fold-change of >5 and <0.2 with no detected flag in either the denominator or the numerator was used as criteria to select the differentially expressed miRNAs. The predicted genes from the changes in miRNA expression and miRNA–mRNA interaction was analyzed by MiRTarBase 7.0 (National Chiao Tung University, Hsinchu, Taiwan).

### 4.6. Statistical Analyses

The data in Figure 1 are expressed as the mean ± standard deviation. Statistical significance was assessed using the unpaired Student’s *t*-test (Microsoft Excel 2013). Results with *p*-values of less than 0.05 were considered statistically significant.

## 5. Conclusions

To the best of our knowledge, this study is the first report of a comprehensive analysis of miRNA expression in different ocular component tissues of LIM mice. Further analyses, such as a cluster analysis or a gene ontology (GO) analysis, are required for a full understanding of the function of differentially expressed miRNAs in different tissues. A comparison to changes in mRNA expression in myopia progression is also important to reveal interactions between miRNA and mRNA. Based on the findings in this study, miRNAs that play a critical role in myopia development and progression may be found and adopted for clinical use as therapeutic targets or diagnostic tools in the future.

## Figures and Tables

**Figure 1 ijms-20-03629-f001:**
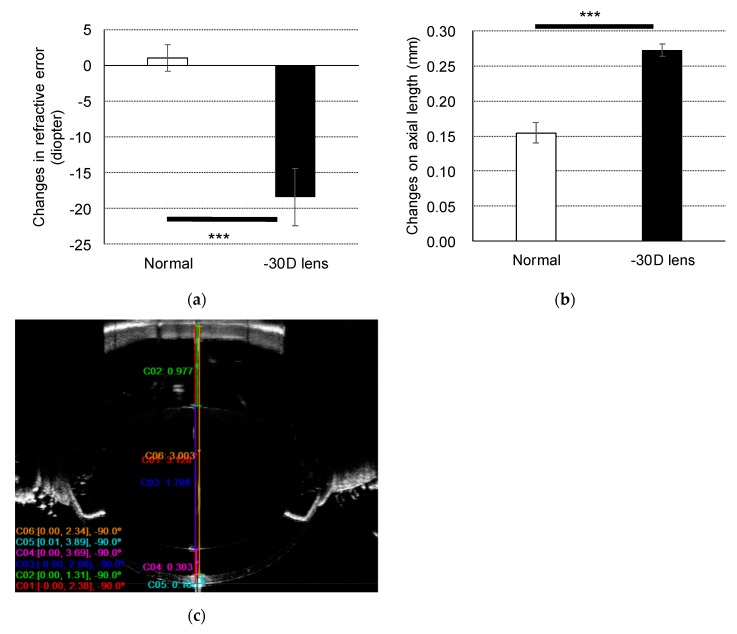
Myopia induction in mice using a negative 30 diopter (D) lens. Changes in (**a**) refraction and (**b**) axial length over three weeks are shown as a comparison between normal and negative 30 D lens-wearing eyes (*n* = 3). Data are presented as the mean ± SD. *** *p* < 0.001; Student’s *t*-test. (**c**) A representative optical coherent tomography (OCT) image showing each ocular component.

**Figure 2 ijms-20-03629-f002:**
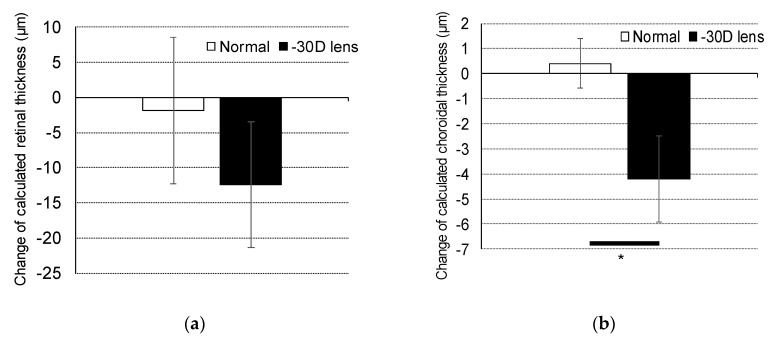
Changes in retina and choroid thickness during the period of lens-induced myopia. The changes in (**a**) retina and (**b**) choroid thickness over a three-week-period are shown as a comparison between normal and negative 30 D lens-wearing eyes (*n* = 3). Data are presented as the mean ± SD. * *p* < 0.05; Student’s *t*-test.

**Figure 3 ijms-20-03629-f003:**
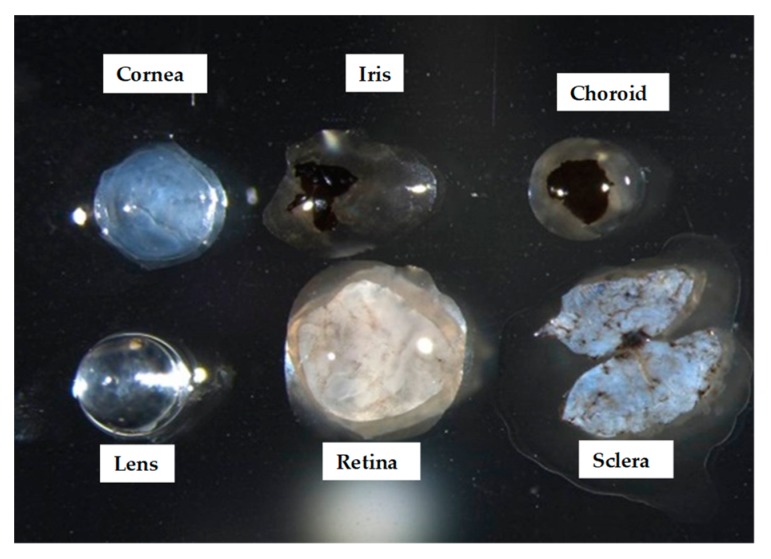
Ocular component tissues. The individual components of the eyeball are shown as separated into the cornea, iris, lens, retina, retinal pigment epithelium (RPE)/choroid, and sclera. The RPE/choroid and the sclera components were removed as delicately as possible using surgical scalpels.

**Figure 4 ijms-20-03629-f004:**
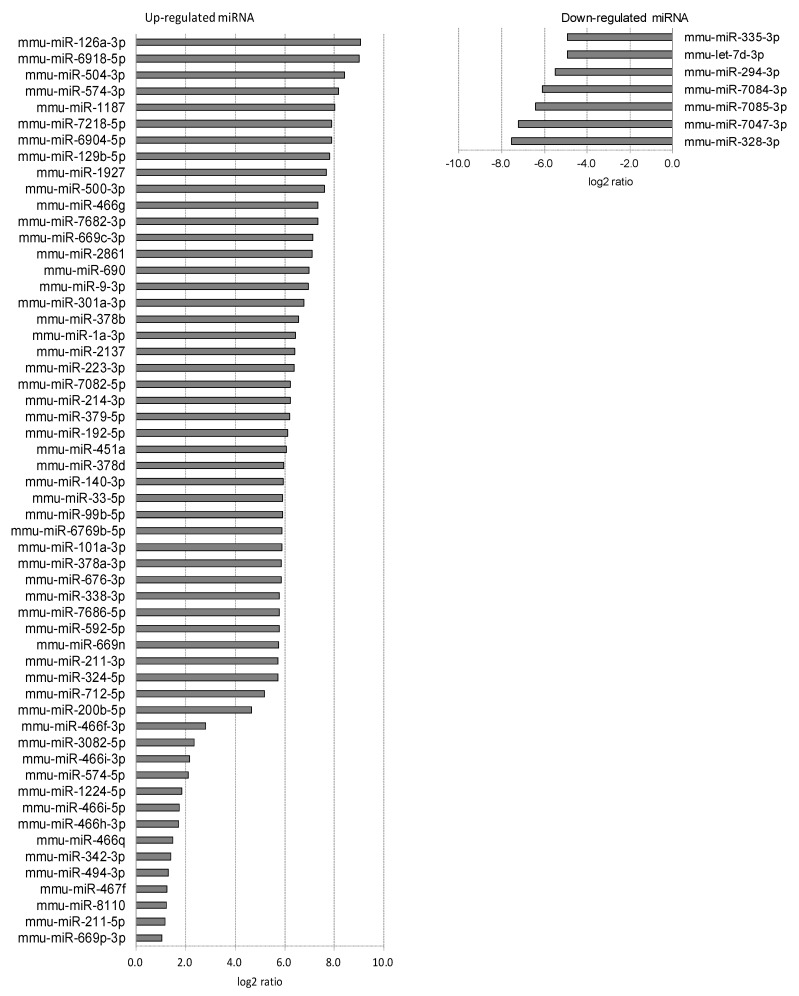
Corneal miRNAs affected by myopia induction. The corneal miRNAs whose expression was affected by myopia induction are listed as upregulated (**left**) and downregulated (**right**).

**Figure 5 ijms-20-03629-f005:**
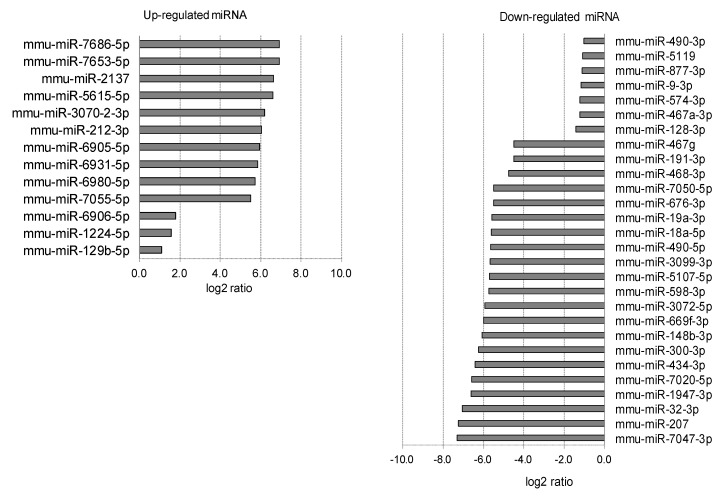
Iris miRNAs affected by myopia induction. The iris miRNAs whose expression was affected by myopia induction are listed as upregulated (**left**) and downregulated (**right**).

**Figure 6 ijms-20-03629-f006:**
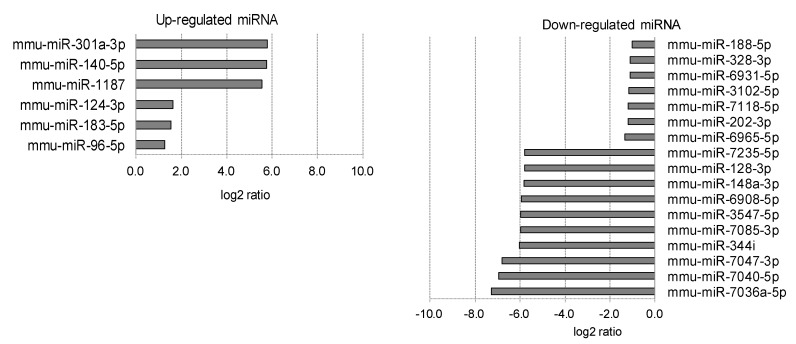
Lens miRNAs affected by myopia induction. The lens miRNAs whose expression was affected by myopia induction are listed as upregulated (**left**) and downregulated (**right**).

**Figure 7 ijms-20-03629-f007:**
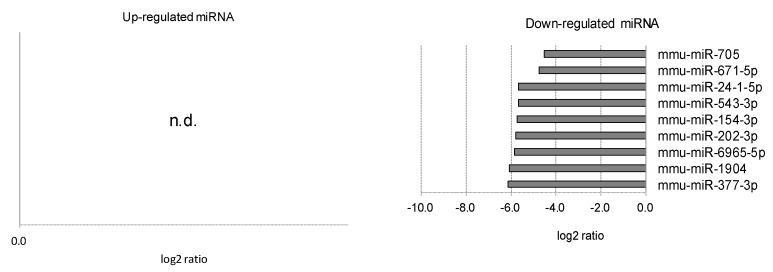
Retinal miRNAs affected by myopia induction. The retinal miRNAs whose expression was affected by myopia induction are listed as upregulated (**left**) and downregulated (**right**). Note that no upregulated miRNAs were detected (n.d.).

**Figure 8 ijms-20-03629-f008:**
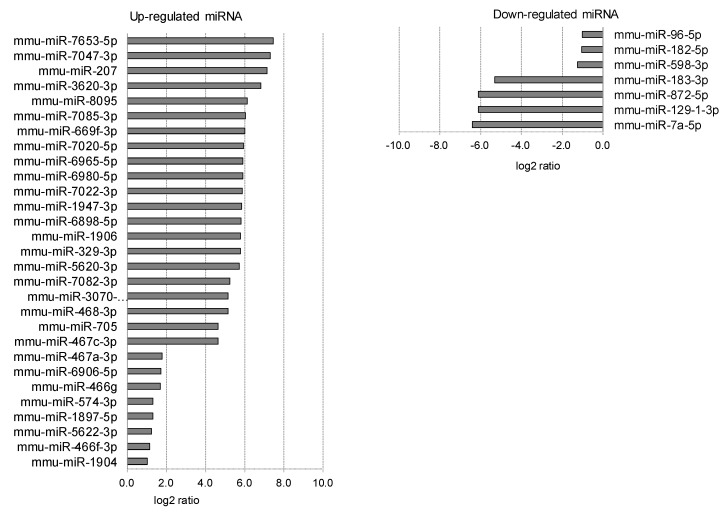
Retinal pigment epithelium (RPE)/choroidal miRNAs affected by myopia induction. The RPE/choroidal miRNAs whose expression was affected by myopia induction are listed as upregulated (**left**) and downregulated (**right**).

**Figure 9 ijms-20-03629-f009:**
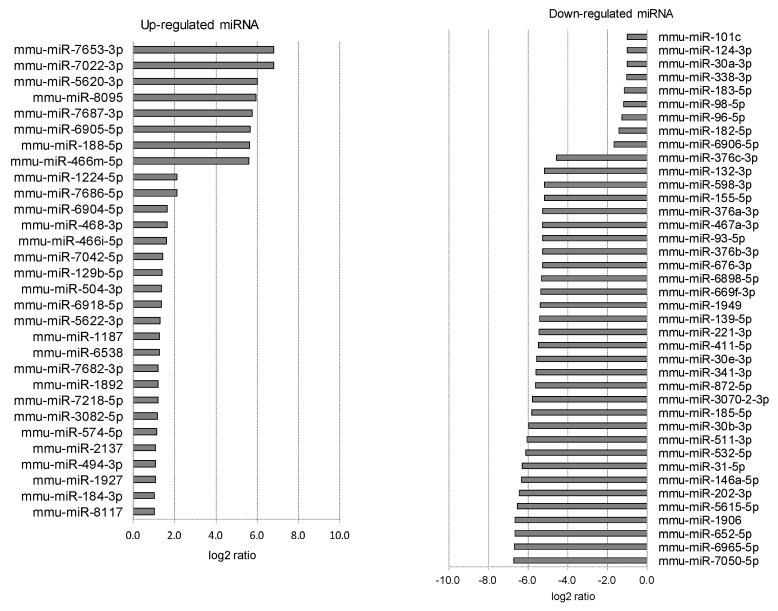
Scleral miRNAs affected by myopia induction. The scleral miRNAs whose expression was affected by myopia induction are listed as upregulated (**left**) and downregulated (**right**).

**Table 1 ijms-20-03629-t001:** The numbers of differentially expressed microRNAs (miRNAs) affected by myopia induction in the individual ocular components.

	Combination				2 Times Over or 0.5 Times Less *^1^	5 Times Over *^2^ or 0.2 Times Less *^3^	Total
1	Cornea	Numerator	Lens-induced	**Up**	**14**	**42**	**56**
		Denominator	Normal	Down	0	7	7
2	Iris	Numerator	Lens-induced	**Up**	**3**	**10**	**13**
		Denominator	Normal	Down	7	21	28
3	Lens	Numerator	Lens-induced	**Up**	**3**	**3**	**6**
		Denominator	Normal	Down	7	10	17
4	Retina	Numerator	Lens-induced	**Up**	**0**	**0**	**0**
		Denominator	Normal	Down	0	9	9
5	RPE/choroid	Numerator	Lens-induced	**Up**	**8**	**21**	**29**
		Denominator	Normal	Down	3	4	7
6	Sclera	Numerator	Lens-induced	**Up**	**22**	**8**	**30**
		Denominator	Normal	Down	9	31	40

*^1^ Detected flag indicated and expression ratio 2 times over or 0.5 times less both groups. *^2^ Not detected flag was indicated in numerator, but detected flag in denominator and expression ratio of 5 times over. *^3^ Not detected flag was indicated in denominator, but detected flag in numerator and expression ratio of 0.2 times less was showed.

**Table 2 ijms-20-03629-t002:** Overlapping expression changes in miRNAs in corneal and other ocular tissues.

Cornea			
**Up** **Each Eye-Tissues**		Down Each Eye-Tissues	
**up**	down	**up**	down
**Iris**		**Iris**	
mmu-miR-7686-5p	mmu-miR-676-3p	-	mmu-miR-7047-3p
mmu-miR-2137	mmu-miR-574-3p	Lens	
mmu-miR-1224-5p	mmu-miR-9-3p	-	mmu-miR-7047-3p
mmu-miR-129b-5p			mmu-miR-7085-3p
**Lens**			mmu-miR-328-3p
mmu-miR-301a-3p	-	**Retina**	
mmu-miR-1187		-	-
**Retina**		**RPE/choroid**	
-	-	mmu-miR-7085-3p	-
**RPE/choroid**		mmu-miR-7047-3p	
mmu-miR-466f-3p	-	**Sclera**	
mmu-miR-574-3p		-	-
mmu-miR-466g			
**Sclera**			
mmu-miR-1927	mmu-miR-676-3p mmu-miR-338-3p		
mmu-miR-494-3p			
mmu-miR-2137			
mmu-miR-574-5p			
mmu-miR-3082-5p			
mmu-miR-7218-5p			
mmu-miR-7682-3p			
mmu-miR-1187			
mmu-miR-6918-5p			
mmu-miR-504-3p			
mmu-miR-129b-5p			
mmu-miR-466i-5p			
mmu-miR-6904-5p			
mmu-miR-7686-5p			
mmu-miR-1224-5p			

**Table 3 ijms-20-03629-t003:** Overlapping expression changes in miRNAs in iris and other ocular tissues.

Iris			
**Up** **Each Eye-Tissues**		Down Each Eye-Tissues	
**up**	down	**up**	down
**Lens**		**Lens**	
	mmu-miR-6931-5p	-	mmu-miR-128-3p
**Retina**		**Retina**	
-	-	-	-
**RPE/choroid**		**RPE/choroid**	
mmu-miR-7653-5p	-	mmu-miR-207	mmu-miR-598-3p
mmu-miR-3070-2-3p		mmu-miR-1947-3p	
mmu-miR-6980-5p		mmu-miR-7020-5p	
mmu-miR-6906-5p		mmu-miR-669f-3p	
**Sclera**		mmu-miR-468-3p	
mmu-miR-6905-5p	mmu-miR-5615-5p	mmu-miR-467a-3p	
	mmu-miR-3070-2-3p	**Sclera**	
	mmu-miR-6906-5p	mmu-miR-468-3p	mmu-miR-598-3p
			mmu-miR-7050-5p
			mmu-miR-467a-3p
			mmu-miR-669f-3p

**Table 4 ijms-20-03629-t004:** Overlapping expression changes in miRNAs in lens and other ocular tissues.

Lens			
**Up** **Each Eye-Tissues**		Down Each Eye-Tissues	
**up**	down	**up**	down
**Retina**		**Retina**	
-	-	-	mmu-miR-6965-5p
**RPE/choroid**			mmu-miR-202-3p
mmu-miR-96-5p	-	**RPE/choroid**	
**Sclera**		mmu-miR-6965-5p	-
-	mmu-miR-96-5p	**Sclera**	
	mmu-miR-183-5p	mmu-miR-188-5p	mmu-miR-6965-5p
	mmu-miR-124-3p		mmu-miR-202-3p

**Table 5 ijms-20-03629-t005:** Overlapping expression changes in miRNAs in retina and other ocular tissues.

Retina			
**Up** **Each Eye-Tissues**		DownEach Eye-Tissues	
**up**	down	**up**	down
**RPE/choroid**		**RPE/choroid**	
-	-	mmu-miR-1904	-
**Sclera**		mmu-miR-705	
-	-	**Sclera**	
		-	-

**Table 6 ijms-20-03629-t006:** Overlapping expression changes in miRNAs in RPE/choroid and other ocular tissues.

RPE/Choroid			
**Up** **Each Eye-Tissues**		Down Each Eye-Tissues	
**up**	down	**up**	down
**Sclera**		**Sclera**	
mmu-miR-5622-3p	mmu-miR-1906 mmu-miR-6898-5p	-	mmu-miR-872-5p
mmu-miR-8095			mmu-miR-182-5p
mmu-miR-5620-3p			
mmu-miR-7022-3p			

**Table 7 ijms-20-03629-t007:** Overlapping expression changes in miRNAs between more than three types of ocular tissue. Note that grey highlights indicate downregulated miRNAs.

miRNAs	Overlapping Tissues			
mmu-miR-1187	Cornea	Lens	Sclera	
mmu-miR-1224-5p	Cornea	Iris	Sclera	
mmu-miR-129b-5p	Cornea	Iris	Sclera	
mmu-miR-202-3p	Lens	Retina	Sclera	
mmu-miR-2137	Cornea	Iris	Sclera	
mmu-miR-3070-2-3p	Iris	RPE/choroid	Sclera	
mmu-miR-467a-3p	Iris	RPE/choroid	Sclera	
mmu-miR-468-3p	Iris	RPE/choroid	Sclera	
mmu-miR-574-3p	Cornea	Iris	RPE/choroid	
mmu-miR-598-3p	Iris	RPE/choroid	Sclera	
mmu-miR-669f-3p	Iris	RPE/choroid	Sclera	
mmu-miR-676-3p	Cornea	Iris	Sclera	
mmu-miR-6906-5p	Iris	RPE/choroid	Sclera	
mmu-miR-6965-5p	Lens	RPE/choroid	Sclera	
mmu-miR-7047-3p	Cornea	Iris	Lens	RPE/choroid
mmu-miR-7085-3p	Cornea	Lens	RPE/choroid	
mmu-miR-7686-5p	Cornea	Iris	Sclera	
mmu-miR-96-5p	Lens	RPE/choroid	Sclera	

**Table 8 ijms-20-03629-t008:** The target genes predicted from the change in miRNA expression in the cornea.

Cornea			
Upregulated miRNAs	Target gene	Downregulated miRNAs	Target gene
mmu-miR-211-5p	PPM1K	mmu-miR-328-3p	BACE1
mmu-miR-494-3p	**FGFR2**, ROCK1	mmu-miR-294-3p	CDKN1A
mmu-miR-200b-5p	ARHGDIA		
mmu-miR-669n	SENP6		
mmu-miR-338-3p	**FGFR2**, Runx2, TNFSF11		
mmu-miR-378a-3p	Nrf1, GLI3, MAPK1		
mmu-miR-101a-3p	COX2, DUSP1, ACKR3, NeuroD1		
mmu-miR-99b-5p	MFGE8		
mmu-miR-33-5p	ABCG1, ABCA1, NRIP1, CCL2, HMGA2		
mmu-miR-451a	MYC, YWHAZ, ANKRD46		
mmu-miR-192-5p	H3F3BH		
mmu-miR-214-3p	POU4F2, SP7, DIO3, Atg12, **PTEN**		
mmu-miR-223-3p	MEF2C, STAT3, GRIA2, GRIN2B, FOXO3, IGF1R, LIF, IL6		
mmu-miR-1a-3p	CALM2, CALM1, MEF2A, IGF1, HAND2, IRX5, GJA1, KLF4, CDC42, TLX2, ANXA5, MKL1, SMARCD2, CLCN3, SMARCB1, MAP4K3, **FZD7**, MEOX2, NFAT5, RARB, SH3BGR1, MYOCD, ADAR, BDNF		
mmu-miR-301a-3p	PIAS3, SOCS5, IRF1, **PTEN**		
mmu-miR-9-3p	SCN2B		
mmu-miR-690	VCAN, CTNNB1		
mmu-miR-126a-3p	PIK3R2, IRS1, ITGA11, ERRFI1, **FZD7**		

**Table 9 ijms-20-03629-t009:** The target genes predicted from the change in miRNA expression in the iris.

Iris			
Upregulated miRNAs	Target gene	Downregulated miRNAs	Target gene
mmu-miR-212-3p	MMP9, HMGB1	mmu-miR-434-3p	VCAN, CTNNB1, EIF5A
		mmu-miR-148b-3p	CAMK2A, DNMT1
		mmu-miR-18a-5p	HSF2, HIF1A
		mmu-miR-19a-3p	ZFPM2, PTEN, TNF, FZD4, LRP6
		mmu-miR-468-3p	HELLS
		mmu-miR-467g	RUNX2
		mmu-miR-128-3p	POU4F2, NF1, PPARA, RUNX1, PAX3, ABHD5, MAPK14, DCX
		mmu-miR-9-3p	SCN2B
		mmu-miR-877-3p	SMAD7
		mmu-miR-490-3p	NANOG

**Table 10 ijms-20-03629-t010:** The target genes predicted from the change in miRNA expression in the lens.

Lens			
Upregulated miRNAs	Target gene	Downregulated miRNAs	Target gene
mmu-miR-301a-3p	PIAS3, SOCS5, IRF1, PTEN	mmu-miR-148a-3p	CAMK2A, KDM6B, ROCK1, MET
mmu-miR-140-5p	HDAC4, ASP1, TGFBR1, WNT11, OSTM1	mmu-miR-128-3p	POU4F2, NF1, PPARA, RUNX1, PAX3, ABHD5, MAPK14, DCX
mmu-miR-124-3p	ITGB1, FOXA2, SYCP1, PTBP1, CTDSP1	mmu-miR-328-3p	BACE1
mmu-miR-183-5p	ZEB2, LRP6		
mmu-miR-96-5p	CLIC5, INSIG2, AKT1S1		

**Table 11 ijms-20-03629-t011:** The target genes predicted from the change in miRNA expression in the RPE/choroid.

RPE/choroid			
Upregulated miRNAs	Target gene	Downregulated miRNAs	Target gene
mmu-miR-468-3p	HELLS	mmu-miR-7a-5p	HELLS, EIF4E, RPS6KB1, MAPKAP1, MKNK2, MKNK1, Sp1, PARP1, NLRP3, HERPUD2, MYRIP, PAX6, KLF4, PTK2
mmu-miR-329-3p	DLK1		
		mmu-miR-182-5p	**CLIC5**, FBXW7, MYOD1, TLR4
		mmu-miR-96-5p	**CLIC5**, INSIG2, AKT1S1

**Table 12 ijms-20-03629-t012:** The target genes predicted from the change in miRNA expression in the sclera.

Sclera			
Upregulated miRNAs	Target gene	Downregulated miRNAs	Target gene
mmu-miR-184-3p	FZD4, NUMBL, SLC25A22	mmu-miR-146a-5p	IRAK2, TRAF6, **VSIVGP2**, **NOTCH1**, MED1, RELB, IRAK1, MAP1B, RNF11, **WNT1**, WNT5A, EGR1, PPP3R2, TGIF1, CAMK2D
mmu-miR-494-3p	FGFR2, ROCK1		
mmu-miR-468-3p	HELLS		
		mmu-miR-31-5p	HIF1A, PDGFB, FZD4, TGFB2, VAV3
		mmu-miR-30b-3p	PPIF
		mmu-miR-185-5p	KDM6B, VCAN, CTNNB1
		mmu-miR-30e-3p	SPATA19, NPFFR2, MRPS30
		mmu-miR-221-3p	DDIT4, KIT, ARNT
		mmu-miR-139-5p	FOXO1, **NOTCH1**, IRS1
		mmu-miR-376b-3p	NFKBIZ, **STAT3**, HOXD10
		mmu-miR-93-5p	**STAT3**, **VSIVGP2**, SQSTM1, NFE2L2, BMPR2
		mmu-miR-376a-3p	PCNA
		mmu-miR-155-5p	RHOA, RHEB, SOCS1, AICDA, PEA15A, MAF, SPI1, FGF7, INPP5D, FOS, PMAIP1, CLDN1, HDAC4, NR1H3, CISH, GSK3B, CSNK1A1
		mmu-miR-132-3p	MECP2, EP300, KDM5A, BTG2, PAIP2, MMP9, NR4A2, ARHGAP32, MAPT, SOX4
		mmu-miR-182-5p	**CLIC5**, FBXW7, MYOD1, TLR4
		mmu-miR-96-5p	**CLIC5**, INSIG2, AKT1S1
		mmu-miR-98-5p	ACVR1B, MMP11, IL6, **WNT1**
		mmu-miR-183-5p	ZEB2, LRP6
		mmu-miR-338-3p	FGFR2, RUNX2, TNFSF11
		mmu-miR-124-3p	ITGB1, FOXA2, SYCP1, PTBP1, PTBP2, CTDSP1, CEBPA, DLX2, SOX9, PTBP2, DLX5, CCNA2, NR3C2, PIM1, CAV1

**Table 13 ijms-20-03629-t013:** The overlapping expression changes in miRNAs found in two ocular tissues and their target genes.

Overlapping Expression Changes of miRNAs in Two Tissues and Target Genes	
**Upregulated in both cornea and lens**	**Target gene**
mmu-miR-301a-3p	FGFR2, RUNX2, TNFSF11
**Upregulated in both cornea and sclera**	
mmu-miR-494-3p	FGFR2, ROCK1
**Downregulated in both cornea and lens**	
mmu-miR-328-3p	BACE1
**Upregulated in cornea and downregulated in iris**	
mmu-miR-9-3p	SCN2B
**Upregulated in cornea and downregulated in sclera**	
mmu-miR-338-3p	PIAS3, SOCS5, IRF1, PTEN
**Downregulated in both iris and lens**	
mmu-miR-128-3p	POU4F2, NF1, PPARA, RUNX1, PAX3, ABHD5, MAPK14, DCX
**Upregulated in lens and downregulated in sclera**	
mmu-miR-124-3p	ITGB1, FOXA2, SYCP1, PTBP1, CTDSP1
mmu-miR-183-5p	ZEB2, LRP6
**Downregulated in both RPE/choroid and sclera**	
mmu-miR-182-5p	CLIC5, FBXW7, MYOD1, TLR4

**Table 14 ijms-20-03629-t014:** The overlapping expression changes in miRNAs found in three ocular tissues and their target genes.

Overlapping Expression Changes of miRNAs in Three Tissues and Target Gene	
**Upregulated in lens and RPE/choroid, downregulated in sclera**	**Target gene**
mmu-miR-96-5p	CLIC5, INSIG2, AKT1S1
**Upregulated in iris, RPE/choroid and sclera**	
mmu-miR-468-3p	HELLS

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
