# Peer review of "Ocular-Component-Specific miRNA Expression in a Murine Model of Lens-Induced Myopia"

_ijms, 2019, doi:10.3390/ijms20153629_

Round 1

Reviewer 1 Report

The manuscript entitled “Ocular component specific miRNA expression in a murine model of lens-induced myopia” by Tanaka et al use different components of mouse lens-induced myopia eyes and did a comprehensive comparation of differentiated expressed miRNAs. The miRNA profile in myopia is interesting. The article is good written. Although differentiated expressed microRNAs were identified, the potential target genes of microRNAs were not found/discussed (not even for prediction), nor future implications were highlighted. The major shortcomings are no validation of these differentiated expressed microRNAs, no validation of potential target genes predicting to interact with microRNAs, and no new insights to the scholarly literature with respect to previously published articles. It’s good to report these data, but the story is not complete.

I have a few major points for the authors to consider revising.    

First, why one to one comparation between different components? This needs to be specified why it’s necessary to compare between two different tissues. Why not one to two or one to pool comparation?

The data reported here are very preliminary. As mentioned by the authors, further analyses such as cluster analysis or GO analysis are required for understanding the function of differentially expressed miRNAs in different tissues. Comparison to mRNA expression changes in myopia induction is also important to reveal interactions between miRNA and mRNA. Other than listing these differentiated miRNAs, it’s suggestable to do a thorough comparison with previous findings, bioinformatics analysis (eg: GO, volcano plot, two-dimensional hierarchical clustering, Venn diagram, MiRTarBase prediction), validations of some miRNAs and their clinical implications/significance.  

Are three mice good enough to drive the statistic power?

Reviewer 2 Report

A well-written paper with facts put forth in a clear and a concise manner. 

Abstract

General Comments: 

Were there any changes observed in retinal and choroidal thickness between occluded and open eyes? If yes pls do include in the abstract. 

Specific Comments: 

Line 16:  Include sample size for each tissue. 

Introduction

Specific Comments: 

Line 27:  Provide Reference.

Line 41: Should include the significance to study the role of MicroRNAs in the progression of myopia.

Results

General comment:

Recommend running Northern blot analysis or quantitative real-time PCR (qPCR) as a validation experiment on target miRNAs. It would be good to include the actual values for axial length and refraction that was taken. Provide SD-Oct images or histology for ocular component measurements to see differences in retinal, choroidal and scleral thickness between lens-induced eyes and control eyes. 

Too small sample sizes used for whole experiments. What was the rationale to choose n = 3 per group? Have you utilized non-occluded eyes (contra-lateral eyes) as a control or naïve eyes as a control sample for miRNA experiments?

Specific Comments:

Figure 1: Current sample size is not convincing to accept the significance level, P < 0.001; n = 3.

Figure 2: The sclera tissue looks contaminated with choroid. How did you prepare the scleral samples for further analysis? Was RPE separated or included with choroid? In that case, you have to specify RPE/choroid complex.

Discussion

General comment

Discuss on species difference in differentially expressed miRNAs and include the significance of miRNAs expression in the progression of myopia.

Methods

General comment

Specify the gender of mice. Did you notice any gender difference in miRNA expression?

Specific Comments:

Line 149: What does authors mean by normal group? Are they different set of n = 3 mice? How many eyes were used as control sample for miRNA run?

Line 153: Specify what overdose of anesthesia was used for euthanasia? Specify the anesthetic drugs used. 

Line 153: All right eyes of -30D pooled as 1 sample for RNA extraction? Which means 1 sample per tissue… and how many run was conducted per sample?

Line 154: Control group or normal group?  And how many eyes? Be specific throughout the manuscript.

Round 2

Reviewer 1 Report

It’s not a bad idea to report the miRNA profile of ocular components in a murine model of myopia, and it would be a super-good reference for the future related studies. I have to say, I am not impressed by the discussion at all. I can’t see the points/initial motivation the authors did this study, the significance and how the findings might change the field. A thorough discussion is invited. And I suggest highlighting how the readers could utilize your discrete and overlapping miRNA data for their future applications.

Author Response

 Reviewer #1

 It’s not a bad idea to report the miRNA profile of ocular components in a murine model of myopia, and it would be a super-good reference for the future related studies. I have to say, I am not impressed by the discussion at all. I can’t see the points/initial motivation the authors did this study, the significance and how the findings might change the field. A thorough discussion is invited. And I suggest highlighting how the readers could utilize your discrete and overlapping miRNA data for their future applications.

Thank you for your constructive suggestion. We added further descriptions to discuss the significance of overlapping miRNA data and their future applications as below:

3. Discussion

It has been reported that miR-200a/b/c expression overlaps in a range of tissues with a tubular structure, including kidney tissue (proximal tubule and collecting duct), lung tissue, pancreas tissue (duct cells), small intestine tissue (intestinal villus), bile duct tissue, and exocrine gland tissue (duct cells). Furthermore, miR-200a/b/c expression was found to be increased in plasma from the site of an acute kidney injury, suggesting that miR-200a/b/c may be used as a biomarker for kidney and other tubular structure organ injury [31]. In the current study, we found overlapping changes in miRNA expression in two and three types of ocular tissue (Tables 2–7). These individual ocular components are in close proximity to and functionally connected with each other. Thus, we suggest that overlapping changes in miRNA expression among different ocular components can be used as myopic diagnosis markers.

In a previous study, eight miRNAs were found to be upregulated and to overlap with retinas and whole eyeballs in a murine model of form-deprivation myopia. The authors screened out 1805 target genes for the eight differentially expressed miRNAs, including MAPK-10 [32]. In the present study, we also found a number of overlapping miRNAs in individual ocular components together with predicted target genes (Tables 13 and 14). Although these new target genes were found to not exactly correspond to previous reports, we speculate that these genes may be important factors in the suppression or acceleration of myopia progression.

Added references:

31. Kito, N.; Endo, K.; Ikesue, M.; Weng, H.; Iwai, N. miRNA Profiles of Tubular Cells: Diagnosis of Kidney Injury. Biomed Res. Int. 2015, 2015, 465479.

32. Mei, F.; Wang, J.; Chen, Z.; Yuan, Z. Potentially Important MicroRNAs in Form-Deprivation Myopia Revealed by Bioinformatics Analysis of MicroRNA Profiling. Ophthalmic Res. 2017, 57, 186–193.

Reviewer 2 Report

No further comments.

Author Response

Reviewer #2

 No further comments.

Thank you very much for giving us the opportunity to strengthen our manuscript with your initial valuable comments and queries.

Round 3

Reviewer 1 Report

The article is improved and I recommend acceptance.